# Relationship Between Dietary Inflammatory Index, Diets, and Cardiovascular Medication

**DOI:** 10.3390/nu17091570

**Published:** 2025-05-02

**Authors:** Teresa Lopez de Coca, Pablo Maya, Victoria Villagrasa, Lucrecia Moreno

**Affiliations:** 1Cátedra DeCo MICOF-CEU UCH, Universidad Cardenal Herrera-CEU, CEU Universities, 46115 Valencia, Spain; teresa.lopezperez@uchceu.es (T.L.d.C.); pablo.maya@alumnos.uchceu.es (P.M.); vvilla@uchceu.es (V.V.); 2Department of Pharmacy, Universidad Cardenal Herrera-CEU, CEU Universities, 46115 Valencia, Spain

**Keywords:** anti-inflammatory, cardiovascular risk, diets, hypertension, nutrition, diets

## Abstract

Cardiovascular (CV) diseases remain a leading global health challenge, being influenced by diet and systemic inflammation. Adherence to healthy dietary patterns, such as the Mediterranean (MED), Dietary Approaches to Stop Hypertension (DASH), and Anti-inflammatory (AnMED) diets, may reduce the CV risk. Background/Objectives: We aimed to evaluate the association between the adherence to healthy dietary patterns and CV treatments. Methods: This cross-sectional study was conducted in the Valencian Community, Spain. Nutritional data were collected using a food frequency questionnaire to assess the adherence to MED, DASH, and AnMED dietary patterns. Statistical analyses, including Kruskal–Wallis tests and linear regression models, evaluated dietary adherence, nutrient intake, the Dietary Inflammatory Index (DII), and medication use. Results: Of 468 participants initially recruited, were included in the final analysis after applying inclusion and exclusion criteria (88.48% female, mean age: 66.16 ± 9.59 years). A significant association was observed between the DII and antihypertensive use (*p*-value < 0.001), with higher DII scores correlating with increased antihypertensive consumption. Among dietary patterns, the AnMED diet exhibited the strongest association with the DII (*p*-adjust < 0.001). Predictive modeling revealed a 14.28% increase in antihypertensive use per unit rise in the DII. The AnMED diet was the only pattern significantly linked to improved micronutrient intake, including calcium, magnesium, sodium, and potassium. Conclusions: The DII is a useful tool for assessing the inflammatory potential of diets. Diets with lower DII scores, such as the AnMED diet, may reduce systemic inflammation and improve CV health. Adherence to the AnMED diet may lower blood pressure and reduce reliance on antihypertensive medications, supporting anti-inflammatory dietary patterns for CV disease prevention and management.

## 1. Introduction

Hypertension (HT) represents the principal risk factor associated with cardiovascular (CV) disease [1]. It is estimated that approximately 1.3 billion individuals globally are afflicted by HT, with its prevalence exhibiting a continuous upward trend, particularly within aging demographics [2]. Chronic low-grade inflammation plays a key role in the pathogenesis of diseases that exhibit shared CV risk factors, including HT, dyslipidaemia, and diabetes mellitus (DM), promotingendothelial dysfunction, oxidative stress, and arterial stiffness, which are key factors in the onset of HT and CV disease [3]. Furthermore, significant inflammatory biomarkers, such as C-reactive protein (CRP), interleukin-6 (IL-6), and tumor necrosis factor-alpha (TNF-α), have been associated with both HT and metabolic disorders [4].

CV pharmacotherapy is indispensable for the management of chronic CV conditions including HT, heart failure, and coronary artery disease. Nevertheless, the application of these medications is accompanied by an array of challenges and adverse effects that may significantly influence patient outcomes such as fatigue, dizziness, electrolyte imbalances, or persistent cough, which can contribute adherence and the quality of life [5,6,7]. 

These medication-related side effects, coupled with the absence of overt symptoms, underscore the necessity for preventive measures as opposed to solely pharmacological interventions. Non-pharmacological strategies have been shown to assist in decreasing the daily dosage of antihypertensive (AHT) medications. Dietary factors significantly influence both the development and management of HT, with critical dietary components encompassing sodium, potassium, alcohol, and overall dietary patterns [8]. For example, excessive sodium consumption has been strongly correlated with elevated blood pressure, whereas increased potassium and calcium intake has been demonstrated to mitigate the hypertensive effects of sodium [9]. Additionally, bioactive compounds present in food, such as polyphenols and omega-3 fatty acids, play a role in the regulation of blood pressure by modulating endothelial function and inflammatory processes [10].

**The Mediterranean diet (MED),** described by the PREDIMED group, has been evidenced to confer advantageous effects on CV health. Its dietary guidelines, which advocate for a high intake of extra virgin olive oil (EVOO), fruits, vegetables, nuts, legumes, and fish, and moderate consumption of red wine, have been associated with a substantial reduction in CV risk and inflammation [11,12]. This dietary pattern is abundant in monounsaturated fats, polyphenols, and dietary fiber, all of which contribute to enhanced lipid profiles, diminished oxidative stress, and improved gut microbiota composition, ultimately leading to a reduction in systemic inflammation [13].

Otherwise, **the Dietary Approaches to Stop HT (DASH)** regimen, developed by the National Institutes of Health, exhibits pronounced efficacy in reducing blood pressure levels. This dietary protocol underscores the consumption of fruits, vegetables, whole grains, and low-fat dairy products while concurrently constraining saturated fat, total fat, cholesterol, and sodium intake and promoting the augmentation of potassium, magnesium, and calcium-rich food sources [14]. Empirical investigations have indicated that adherence to the DASH dietary framework correlates with substantial decreases in both systolic and diastolic blood pressure, irrespective of weight reduction, thereby establishing it as a pivotal dietary intervention for the management of HT [15].

**The Anti-inflammatory diet (AnMED)**, as described by Sala-Climent et al., is the most restrictive among those studied, and is characterized by the inclusion of foods that attenuate inflammation, such as fruits and vegetables, whole grains, nuts, legumes, and plant-derived proteins, while, unlike the MED diet, it restricts or does not explicitly endorse pro-inflammatory foodstuffs such as red and processed meats, refined carbohydrates, saturated fats, dairy products, sugar, pastries or cakes, and alcohol. Furthermore, it accentuates the significance of food diversity and variability. This nutritional paradigm has been demonstrated to possess a significant association with CV health [16,17,18,19]. Importantly, these three dietary models differ in their inflammatory potential, which can be evaluated using the Dietary Inflammatory Index (DII). Novel research indicates that dietary regimens endowed with anti-inflammatory properties may assume a critical role in the prevention of CV disease by diminishing pro-inflammatory cytokines and modulating immune system responses [20]. Table 1 presents the differences between the three dietary models.

Adherence to healthy dietary patterns contributes to CV health and a diminished risk of chronic diseases via a multitude of mechanisms, including reductions in systemic inflammation, the enhancement of endothelial function, the mitigation of oxidative stress, the modulation of lipid profiles, the improvement of insulin sensitivity, and the deceleration of telomere attrition, impacting metabolic syndrome and neuroinflammation. Collectively, these synergistic effects emphasize the critical importance of such dietary patterns in the prevention and management of CV diseases [16,21,22].

In summary, although all three dietary models contribute to CV health, their mechanisms of action differ. The DASH diet is primarily focused on lowering blood pressure through reducing sodium and saturated fats and increases in potassium, magnesium, and calcium intakes. The traditional MED diet exerts its effects through the high intake of EVOO, polyphenols, and omega-3-rich fish, enhancing lipid profiles, antioxidant status, and gut microbiota. The AnMED diet is more restrictive and emphasizes an anti-inflammatory profile by excluding red meats, saturated fats, added sugars, and alcohol while promoting plant-based diversity and specific food groups with antioxidant and immunomodulatory properties, such as turmeric. These mechanistic distinctions may influence their respective inflammatory potential and impact on cardiometabolic outcomes. As such, their varying levels of adherence may result in different DII scores, with the AnMED diet generally associated with lower (more favorable) DII values.

The DII, described by Shivappa et al., serves as a metric designed to assess the inflammatory potential inherent within an individual’s dietary intake. Elevated DII scores correlate with increased cardiometabolic risk factors, including adiposity, waist circumference, inflammatory biomarkers, and CV disease [23]. High DII scores are indicative of a reduced consumption of anti-inflammatory foods such as fruits, vegetables, and fish, alongside a heightened intake of pro-inflammatory foods including red meat and processed products [24]. Diets characterized by lower DII scores, exemplified by the MED diet, are associated with diminished inflammation and a decreased risk of chronic diseases [25], which is achieved through the enhancement of healthy cardiometabolic profiles, including elevated HDL cholesterol levels and improved endothelial function [26]. Furthermore, contemporary findings suggest that the DII is independently correlated with increased blood pressure and arterial stiffness, further underscoring its significance in the management of HT [27].

In light of the established correlation between dietary habits, inflammatory responses, and CV health, it is important to investigate whether compliance with dietary patterns exhibiting anti-inflammatory potentials is adjuvant to pharmacological interventions in individuals diagnosed with HT. The primary aim of this research was to evaluate the association between the consumption of AHT drugs and the DII of their respective dietary patterns within a Spanish demographic. More precisely, this investigation sought to determine whether individuals exhibiting a higher degree of adherence to anti-inflammatory dietary regimens (characterized by lower DII scores) need diminished dosages or a decreased frequency of AHT medication in comparison to those adhering to more pro-inflammatory dietary practices.

Based on the reviewed literature, we hypothesized that individuals with higher DII scores would be more likely to require antihypertensive pharmacological treatment. Furthermore, we expected that pro-inflammatory dietary patterns may be indirectly associated with a greater likelihood or intensity of AHT drug use.

## 2. Materials and Methods

### 2.1. Study Design

A cross-sectional study was conducted over five months, from April to November 2024. Participants were recruited through community pharmacies and healthcare centers in the Valencian Community (Spain).

The inclusion criteria were being at least 50 years old and being willing and able to provide written informed consent. Exclusion criteria included those individuals with any diagnosis of dementia or intellectual disability and/or those experiencing severe sensory deficits (such as blindness or deafness) or physical disability that impaired the ability to participate in the interviews. Additionally, participants without a record of prescribed medications and/or dietary intake data were excluded from the study.

In the present study, the group of CV treatment was considered to involve those patients in chronic treatment with AHT, lipid-lowering treatment (LLT), or antidiabetics (ADM).

### 2.2. Participants and Data Collection

Data were obtained through personal interviews and from the electronic medical records and dispensing data of the participants. This encompassed information pertaining to demographic variables (age and gender), medication regimens, and medical diagnoses. Drug information was documented following, a systematic approach for pharmacotherapeutic follow-up aimed at optimizing medication use, preventing medication-related problems, and improving health outcomes. This method involved the collection and analysis of pharmacotherapeutic data, identification of potential issues, and, if necessary, proposing interventions to ensure treatment efficacy and safety. A database containing all anonymized data was created.

In order to evaluate the nutritional status of the participants, the validated Food Frequency Questionnaire (FFQ) utilized by the PREDIMED group was initially employed to collected data regarding food consumption. The questionnaire comprised 161 items, which participants were asked to complete, indicating their weekly or monthly intake of each item. Daily consumption for each item in grams/day was estimated by multiplying the reported consumption frequency by the average daily intake. The nutritional data and bioactive compounds for each food item were sourced from the Souci–Fachmann–Kraut ‘Food Composition and Nutrition Tables’, 9th edition [28]. Initially, this information was used to calculate the DII for the participants, which was subsequently categorized into low, medium, or high DII based on the 25th, 50th, and 75th percentiles. Subsequently, the data derived from the FFQ were translated into corresponding scores for the three dietary patterns under investigation: the MED [29], DASH [30], and AnMED diets [16]. The AnMED score was recalibrated to a scoring scale ranging from 0 to 14 for the purposes of statistical comparison. For the MED and AnMED dietary models, a score equal to or higher than 9 out of 14 points was classified as high adherence [31]. For the DASH diet, high adherence was a score equal to or greater than 7.5 points out of 10 [30].

The tests used for nutritional assessment consisted of tools specifically designed to measure adherence to various modifications of the MED diet and valuate the frequency of essential dietary components such as fruits, vegetables, nuts, fish, legumes, and olive oil. Additionally, they included inquiries regarding the consumption of less healthy food items, including red meat, processed products, or sugary beverages, among others. These assessments facilitated the identification of dietary patterns and yielded a score that represented the extent to which an individual’s eating habits aligned with a dietary framework, which may have been correlated with a spectrum of health benefits, thereby enhancing overall health.

### 2.3. Statistical Inference

The following procedure describes the calculation of the DII. The Z-score for each nutrient was calculated from the mean and standard deviation of the dietary intake data. These Z-scores were converted to percentiles to minimize the effect of right-skewed distributions. Each percentile was multiplied by two and then reduced by one to produce a symmetrical distribution with a center point of 0 and bounds of −1 and +1. Subsequently, the “Food Parameter Specific DII Score” was computed by multiplying the centered percentile value of each nutrient or food parameter by its corresponding “Overall Inflammatory Response Score”. Finally, the overall “Global DII Score” was determined by summing all the “Food-Specific DII Scores”. DII scores ranged from highly anti-inflammatory (−8.87) to highly pro-inflammatory diets (7.98).

All data on medication use and dietary intake were recorded in an Excel spreadsheet and analyzed using RStudio (v.4.4.2). Linear collinearities between different medications and the DII were assessed, as were their associations with individual food groups. The relationship between food consumption and adherence to the three studied dietary models was examined using the Kruskal–Wallis or Mann–Whitney U test, depending on data distribution. Principal Component Analysis (PCA) was performed using the vegan package to explore food variability based on medication use and the distribution of patients across DII percentiles. A predictive analysis of AHT medication consumption as a function of DII was conducted, applying a statistical model to estimate trends. The association between essential micronutrient intake, including calcium, magnesium, potassium, and sodium, and adherence to the three dietary patterns, was evaluated using linear regression models. Data visualization was carried out with ggplot2, and the dplyr package was used for medication categorization and data processing.

### 2.4. Ethical Approval and Data Protection

The studies involving human participants were reviewed and approved by the Institutional Review Board (IRB) at the CEU Cardenal Herrera University (CEEI24/533; date of approval: 25 April 2024). The participants provided their written informed consent to engage in this study.

Information processing guaranteed both the protection of the data and their security. This data was treated confidentially and lawfully and were used for the purpose for which the participants had been informed. Thus, this work complied with the European General Data Protection Regulation and Organic Law 3/2018 on the Protection of Personal Data and the Guarantee of Digital Rights. The study compiled with the basic principles of the Declaration of Helsinki: respect for the individual (Article 8) and recognition of their right to self-determination and their right to make informed decisions (informed consent, contained in Articles 20, 21, and 22), including participation in research, both at its beginning and throughout the study.

## 3. Results

From an initial recruitment of 468 participants, we excluded those under 50 years of age (*n* = 9), those without dietary records (*n* = 82), and those without medication records (*n* = 73). A total of 304 patients (88.48% female) were finally included in the analysis (Figure 1). Among them, 37 were on AHT, 34 on LLT, and 19 on ADM. Additionally, 13 patients were on concomitant treatment with ADM and LLT, 2 were receiving both AHT and ADM, 34 were treated for both AHT medications and LLT, and 11 were on triple therapy for AHT medications, LLT, and ADM. Moreover, 105 patients were taking other medications but not for CV risk while 49 were not take medication and had no diagnosed pathology. The study population is described in detail in Table 2.

Since there were significant differences in the mean age of the “non-drugs consumption” group compared to the other groups, we examined whether the DII was associated with age. To assess this, we performed a linear regression analysis between the DII and age in the remaining participants. The analysis yielded a *p*-value of 0.164, indicating no significant relationship between diet and age. Despite the observed age differences, this finding supported the inclusion of the “non-drugs consumption” group in the study.

The macronutrient intake of the patients was evaluated (Table 3). Significant differences were observed exclusively between the group with no drug consumption and the groups classified as “CV treatments” and “other drugs”. However, no significant differences were found between the “CV treatments” and “other drugs” groups based on the Dunn test.

### 3.1. Medication Consumption, Diets, and DII

A significant collinearity was observed between AHT use (*p*-value < 0.001) and LLT (*p*-value: 0.002) with an increase in the DII. However, although LLT showed an association, the statistical support was weaker. No significant relationship was found between adherence to the three evaluated diets (MED, DASH, and AnMED) and increased medication use. Only a trend was observed for the AnMED diet and AHT consumption.

Adherence to these diets was significantly associated with the DII in the linear regression model (Figure 2A, MED *p*-value: 0.028; Figure 2B, DASH *p*-value: 0.005; Figure 2C, AnMED *p*-value < 0.001). However, when applying a non-parametric approach (Kruskal–Wallis test with multiple-comparison correction), only the AnMED diet remained statistically significant (*p*-adjusted < 0.001), suggesting a stronger association for this dietary pattern.

Additionally, significant differences were observed in the DII (Figure 2D), berry consumption (Figure 2E), and dietary sodium intake (Figure 2F). The group without medication and chronic conditions exhibited the most favorable results, showing the lowest DII values, the highest berry consumption, and the lowest dietary sodium intake. These findings highlighted the influence of medication and dietary patterns on inflammation and nutrient intake.

### 3.2. Antihypertensive Use and Dietary Intake

In the PCA analysis (Figure 3A), although no significant differences were observed, a clear pattern emerged. Higher AHT consumption was associated with lower dietary variability. Additionally, Kruskal–Wallis tests revealed significant associations for the DII (Figure 3B, *p*-adjusted < 0.001), magnesium (Figure 3C, *p*-adjusted: 0.006), and berry consumption (Figure 3D, *p*-adjusted: 0.033), indicating that these dietary components differed significantly across AHT use categories.

### 3.3. Predicting Antihypertensive Use with DII

To assess the predictive ability of the DII with regards to AHT use, we performed a statistical prediction using a previously fitted model. The objective was to estimate the probability of AHT use for different DII values and provide confidence intervals for the predictions.

A new dataset with hypothetical DII values (−2, −1, 0, 1 and 2) was created to predict the corresponding probabilities of AHT use. As shown in Table 4, each unit increase in the DII was associated with a 14.28% increment in AHT consumption on average. The growth rate of the prediction decreased as DII increased, suggesting that the effect of DII on the response variable was not linear but rather slowed down. This relationship was statistically significant (F-statistic: 24.85, *p*-value < 0.001). However, DII accounted for only 7.6% of the variability in AHT use, indicating that other factors also contributed to this outcome.

The results suggested a positive association between DII and the probability of AHT use. Individuals with a more pro-inflammatory diet (higher DII values) were more likely to use AHT. Specifically, for a DII of −2 (representing an anti-inflammatory diet), the predicted probability of AHT use was 16.81%, whereas for a DII of 2 (pro-inflammatory diet), this probability increased to 73.94%.

These findings reinforced the potential role of dietary inflammation in CV health and highlighted the importance of dietary interventions in managing HT risk.

### 3.4. DII Categorization and Micronutrient Intake

After categorizing DII into percentiles (low ≤ P25, medium (P25-P75), and high ≥ P75), a positive correlation was observed between the improved anti-inflammatory index (lower DII values) and increased magnesium (Figure 4D, *p*- adjusted < 0.001), calcium (Figure 4E, *p*- adjusted < 0.001), potassium (Figure 4F, *p*- adjusted < 0.001), and sodium intakes (Figure 4G, *p*- adjusted: 0.022). Additionally, a significant reduction in AHT use was observed in participants with lower DII scores (Figure 4C, *p*- adjusted < 0.001). In the PCA analysis (Figure 4A), although no significant differences were found, higher pro-inflammatory DII levels appeared to be associated with lower dietary variability.

The collinearity of each class of AHT drugs was also analyzed individually. A significant association was observed between higher DII scores and an increased use of beta-blockers (*p* = 0.005) and angiotensin receptor blockers (ARBs, *p* = 0.015).

### 3.5. Dietary Micronutrient Intake and Modeled Diets

The study analyzed the dietary intake of calcium, magnesium, potassium, and sodium (Table 5).

MED adherence showed no significant association with micronutrient intake.DASH adherence was significantly related to increased sodium (*p*-value: 0.006) and potassium intake (*p*-value: 0.031).AnMED adherence showed a significant relationship with all micronutrients analyzed in a linear model: calcium and magnesium (*p*-value < 0.001), followed by sodium (*p*-value: 0.002) and potassium (*p*-value: 0.039).

## 4. Discussion

The DII is a valuable tool in nutritional research for understanding the role of diet in inflammation and chronic disease development [32]. It can guide dietary interventions aimed to reduce inflammation and preventing chronic conditions [25].

The DII can be used in clinical settings to assess dietary patterns and provide personalized dietary recommendations to reduce inflammation and manage chronic conditions.

Higher DII scores, indicating a more pro-inflammatory diet, are associated with increased CV risk. A meta-analysis has confirmed that a pro-inflammatory diet is associated with a 36% increased risk of CV incidence and mortality [33].

Based on the provided studies, there is evidence suggesting a relationship between a high inflammatory diets and increased statin use, although the connection is not straightforward. A higher level of the DII is associated with higher levels LDL cholesterol in apparently healthy populations [34,35,36]. Consequently, there will also be an increase in the consumption of statin. However, there also seems to be an inverse relationship between the two. There is evidence that statin users may consume more calories and fats, potentially due to a perceived reduction in the need for dietary control once on medication [37]. This behavior could lead to a higher DII.

A recent meta-analysis has reported a significant association between higher DII values and an elevated risk of HT, estimating a 4% increase in HT incidence per one-unit increase in DII (RR = 1.04; 95% CI: 1.00–1.07) [38]. Given that ARBs are commonly prescribed in the treatment of HT, this indirect evidence supports the plausibility of a relationship between a pro-inflammatory diet and the need for AHT medication.

Our predictive model further reinforces this connection. By estimating the probability of AHT use across a range of hypothetical DII values, we observed that each additional DII unit corresponded to an average 14.28% increase in the likelihood of AHT drug use. The association was statistically significant (F-statistic = 24.85, *p* < 0.001), although the model explained only a modest portion of the variance (R^2^ = 0.076), suggesting that other contributing factors were also at play. This modest explanatory power likely reflected the multifactorial nature of antihypertensive drug use, which is influenced by a wide range of variables such as comorbidities, genetic predisposition, lifestyle habits, and access to healthcare. Additionally, the DII, while informative, may not fully account for the complexity of dietary behaviors and their physiological effects.

Moreover, it is important to acknowledge that our study’s cross-sectional design limited the ability to draw causal conclusions. The temporal direction of the association between DII and antihypertensive treatment could not be firmly established, meaning that we cannot exclude the possibility that individuals receiving pharmacological treatment may also have modified their diets in response to their diagnosis.

Notably, the probability of AHT use rose from 16.81% at a DII of –2 (reflecting an anti-inflammatory dietary pattern) to 73.94% at a DII of +2 (indicating a strongly pro-inflammatory diet). These findings underscore the potential influence of dietary inflammation on people with CV risk and pharmacological treatment, highlighting the importance of considering diet as a modifiable factor in hypertension prevention strategies.

Several studies have highlighted the role of pro-inflammatory diets in promoting chronic inflammation and increasing the risk of CV disease [23,39]. Although the existing literature has not established a direct relationship between these dietary patterns and the use of specific AHT drugs, particularly beta-blockers, these conditions often require pharmacological treatment. In line with this, our analysis revealed a significant association between higher DII scores and the increased use of both beta-blockers (*p*-value: 0.005) and ARBs (*p*-value 0.015). Among them were the most prescribed antihypertensive drugs, especially in patients with multiple comorbidities, such as obesity, diabetes, or metabolic syndrome, conditions often accompanied by low-grade chronic inflammation and suboptimal dietary patterns. Given their differing mechanisms of action, it is plausible that their association with higher DII scores reflected not a pharmacological effect per se, but, rather, the inflammatory and dietary profiles of the patients who required them. Although these findings did not confirm causality, they suggest a possible indirect relationship that merits further investigation in future studies.

Considering DM, hypercholesterolemia, and HT as components of CV risk, we must not forget the lack of adherence to nutritional interventions and diets focused on these pathologies.

Despite the well-documented benefits of high adherence to healthy dietary patterns, the proportion of people who achieve such adherence remains relatively low, especially among older adults [40]. Previous research highlights that only 22.3% of patients with type 2 diabetes adhered to the Mediterranean diet and fewer than half engaged in regular physical activity [41]. Similarly, while 83.8% of people with hypercholesterolemia adhered to at least one non-pharmacological measure, only 29.5% followed all the recommended measures, such as weight control, physical activity, and reducing saturated fat intake [42].

The macronutrient intake analysis revealed significant differences between non-drug consumption and receiving CV treatments or other drugs, particularly in total fat and specific fatty acid consumption. The non-drug-consumption group showed a lower total fat intake (51.6%) compared to the CV treatment group (57.6%) and the other-drugs group (54.9%). Despite this, all three groups exceeded the recommended fat intake levels [43]. Notably, saturated fatty acid (SFA) intake was significantly lower in the non-medicated group (13.09 ± 3.32 g) than in the CV treatment (17.95 ± 6.70 g) and other-drugs groups (17.39 ± 6.98 g). Elevated SFA consumption was associated with increased LDL cholesterol levels, a known risk factor for atherosclerosis and CV diseases.

Conversely, the non-drug-consumption group also had lower intakes of MUFA and PUFA, which were cardioprotective but remained within recommended levels. Higher PUFA intake has been inversely linked to coronary heart disease risk. The analysis further highlighted significant differences in specific fatty acid consumption. For instance, lauric acid intake was notably lower in the non-drug-consumption group (0.11 ± 0.13 g) compared to the CV treatment (0.20 ± 0.17 g) and other-drugs groups (0.21 ± 0.19 g). High lauric acid diets have been associated with increased hepatic triglycerides, adipose tissue inflammation, insulin resistance, and liver injury under high-fat dietary contexts [44].

Similarly, oleic acid intake was lower in the non-drug-consumption group (47.66 ± 14.78 g) compared to the CV treatment (65.77 ± 31.59 g) and other-drugs groups (60.84 ± 30.22 g). Oleic acid is widely recognized for its role in reducing LDL cholesterol and improving lipid profiles, as observed in studies on adherence to the Mediterranean diet [45]. However, excessive oleic acid consumption may displace essential fatty acids, such as omega-3 and omega-6, required for balanced cellular, inflammatory, and metabolic functions [46].

Other fatty acids, including myristic, palmitic, and stearic acids, were also consumed in significantly lower amounts by the non-drug-consumption group. For example, the stearic acid intake was 2.71 ± 0.66 g in the non-medicated group compared to 3.66 ± 1.35 g and 3.58 ± 1.38 g in the CV treatment and other-drugs groups, respectively. While palmitic and myristic acids are considered hypercholesterolemic [47,48], stearic acid has a neutral effect on cholesterol levels, underscoring the need for nuanced dietary recommendations [49]. This highlights the complexity of fatty acid profiles and their impact on CV health.

Bearing HT in mind, approximately 60% to 90% of hypertensive patients in Spain do not follow the recommended hygienic–dietary measures. This poor compliance represents a significant barrier to the effective management of HT [50].

In this regard, a lack of adherence to dietary interventions often requires increased reliance on medications to manage chronic diseases [51].

The results of our study regarding the intake of berries and their observed relationship with DII and the consumption of medication for CV risk agree with the numerous studies that have highlighted the anti-inflammatory effects of these fruits and their possible usefulness in reducing the risk of CV disease [52].

Regarding the micronutrient results, we should highlight that the AnMED diet improved sodium, magnesium, potassium, and calcium intake in our patients. Magnesium is essential for electrical, metabolic, and vascular homeostasis, and its deficiency is related to HT, DM, dyslipidaemia, and coronary heart disease [53]. In addition to that, adequate potassium intake can lower blood pressure and improve CV outcomes [54] and consequently reduce CV risk [55]. The dietary pattern of the AnMED diet, which restricts pro-inflammatory foods such as red meats, processed products, and added sugars, alongside its emphasis on dietary diversity and the inclusion of nutrient-dense groups such as blue and white fish, daily nuts, and categorized fruits and vegetables, explains its benefits in essential micronutrient intake. These restrictions reduce the consumption of calorie-dense but nutritionally poor foods while the inclusion of anti-inflammatory and nutrient-rich components enhances the intake and bioavailability of key nutrients such as magnesium, potassium, calcium, and antioxidant compounds. This synergy highlights the potential of the AnMED diet to improve not only nutrient adequacy but also overall metabolic and inflammatory profiles.

On the other hand, the relationship between dietary calcium intake and CV disease is complex, with inconsistent findings. While some studies have suggested that moderate calcium intake may lower the risk of all-cause and CV mortality, others have reported no significant effect [56]. However, we should not forget that it is recommended to maintain a balanced calcium intake (≥1200 mg/day) [43] for general health and CV prevention while avoiding excessive supplementation.

The high content in potassium and calcium with the fruits and vegetables of the MED, DASH, and AnMED diets lead to a natriuretic action [57].

With respect to adherence to the DASH diet and the sodium values [58] obtained from this diet, the results agree with the mean sodium values provided by nutrients in patients following this diet and described in the literature. The DASH-Sodium trial tested various sodium intake levels, including high (3450 mg/day), medium (2300 mg/day), and low (1150 mg/day) sodium diets. The medium sodium intake level of 2300 mg/day is commonly recommended as it balances the need to reduce sodium while maintaining palatability and adherence to the diet. However, there are some results of our work that do not coincide with those reported in the literature [57], which describes that adherence to the DASH diet is generally associated with lower sodium intake. This discrepancy may have arisen from the limitations of the research model employed in this study. The cross-sectional nature of the study did not account for variations in sodium consumption over time or the impact of external factors such as food availability, preparation methods, and cultural influences.

Here are some potential explanations [57,59,60]: individuals may misinterpret the DASH diet guidelines, leading to the inclusion of sodium-rich foods that are not recommended. For example, while the DASH diet encourages the consumption of fresh vegetables, fish, and dairy products, if these are consumed in processed or canned forms, they may contain high levels of sodium. Furthermore, some whole grains, recommended in the DASH diet, contain also moderate amounts of sodium. If consumed in large quantities, these can contribute significantly to the overall sodium intake. In addition, personal preferences and dietary habits can lead to the selection of higher-sodium foods within the DASH diet framework. For example, certain cheeses can contain moderate amounts of sodium or choosing flavored or seasoned versions of recommended foods can inadvertently increase sodium intake.

The DASH-Sodium trial research group conducted a controlled feeding trial to determine the blood pressure effects of sodium restriction alone and in combination with the DASH diet, and the greatest benefit on blood pressure was observed when the low sodium intake was coupled with the DASH diet [61,62].

The Seven Countries Study examined the long-term effects of diet-associated inflammation on mortality over 50 years. Using the energy-adjusted DII, researchers found positive correlations between higher DII scores and increased all-cause and CV disease mortality but not cancer. Stronger correlations were observed after adjusting for socioeconomic status. A higher DII score was also linked to a lower age at death, and these associations remained significant after controlling for blood pressure, cholesterol, and smoking [63].

This study had several limitations that must be acknowledged. Firstly, the cross-sectional design prevented the establishment of causal relationships between DII and the use of AHT drugs. Secondly, the sample size (*n* = 304), although sufficient for detecting statistically significant associations, limits the generalizability of the findings. Thirdly, the use of a convenience sample, with a natural imbalance between older individuals with CV diseases and younger, healthier participants, may have introduced selection bias. However, a linear regression analysis showed no significant relationship between age and the DII (*p* = 0.164), supporting the inclusion of all participant groups. Additionally, the subdivision by dietary adherence and treatment type led to relatively small subgroups, which could have increased the risks of type I errors due to multiple comparisons. These limitations highlight the need for future longitudinal studies with larger, randomized populations to validate our findings.

## 5. Conclusions

The present findings are preliminary and should not be interpreted as sufficient to guide clinical decisions based solely on DII scores.

The DII is a robust widely used tool for evaluating the inflammatory potential of diets and has significant implications for understanding and managing diet-related inflammation and chronic diseases. Encouraging diets with lower DII scores, such as the AnMED diet, may help reduce inflammation and improve CV diseases.

Dietary modifications play a crucial role in managing and preventing CV diseases. These diets, rich in fruits, vegetables, whole grains, and low-fat dairy, and low in sodium, saturated fats, and refined grains, are associated with lower blood pressure and improving overall CV health. Combining these dietary changes with other lifestyle modifications can further enhance their benefits, potentially contributing to a lower probability of requiring AHT treatment in the future.

The consumption of berries is consistently associated with lower levels of inflammation and improved health markers, supporting the idea that a diet rich in berries can contribute to a lower DII.

Rather than replacing pharmacological treatment, anti-inflammatory diets may act as valuable adjuncts in CV disease management by helping reduce systemic inflammation. These findings support dietary recommendations selecting anti-inflammatory foods while limiting pro-inflammatory ones as simple carbohydrates, processed foods, and saturated fats.

## Figures and Tables

**Figure 1 nutrients-17-01570-f001:**
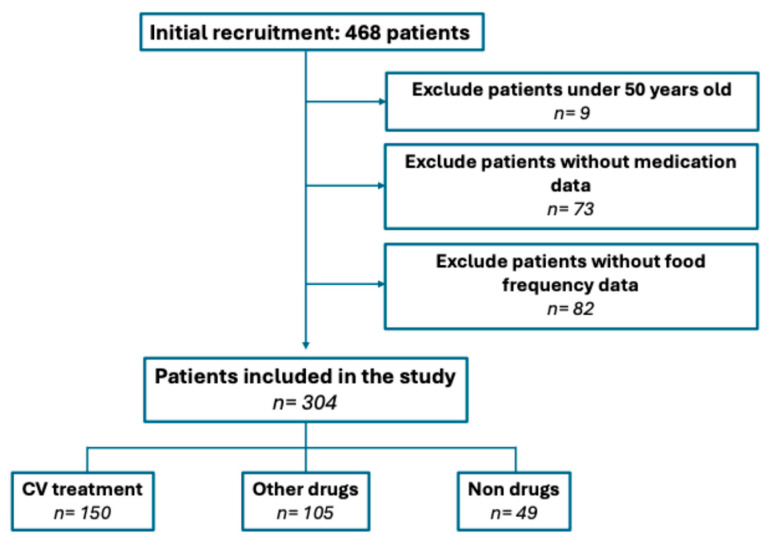
Participant inclusion diagram.

**Figure 2 nutrients-17-01570-f002:**
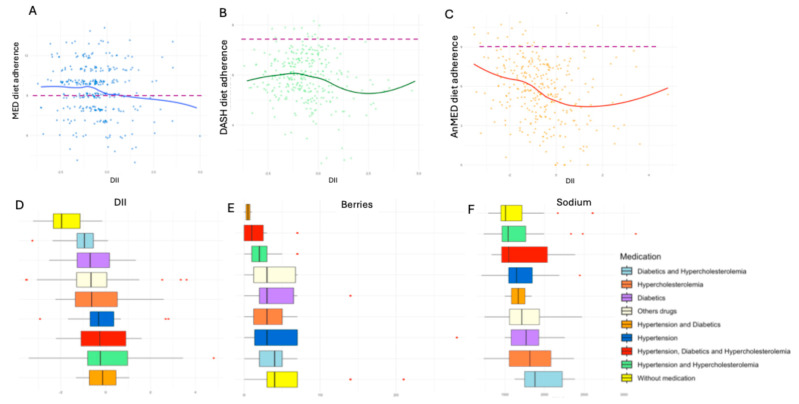
(**A**) Relationship between DII and adherence to MED diet. (**B**) DII and adherence to DASH diet. (**C**) DII and adherence to AnMED diet. The broken line represents the cut-off point between high and low adherence to dietary patterns. (**D**) DII distribution according to the patient’s medication profile. (**E**) Berry consumption in relation to the patient’s medication profile. (**F**) Magnesium intake based on the patient’s medication profile.

**Figure 3 nutrients-17-01570-f003:**
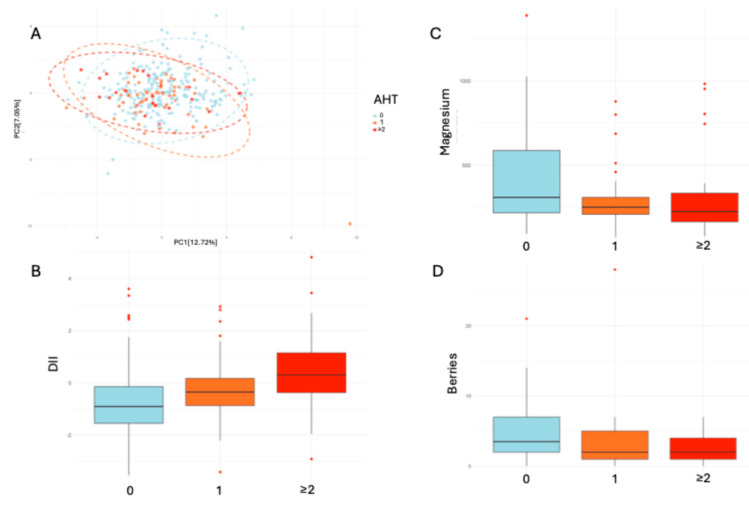
(**A**) PCA of dietary variability based on AHT treatments. (**B**) DII in relation to AHT drug use. (**C**) Magnesium intake in relation to AHT drug use. (**D**) Berry intake in relation to AHT drug use.

**Figure 4 nutrients-17-01570-f004:**
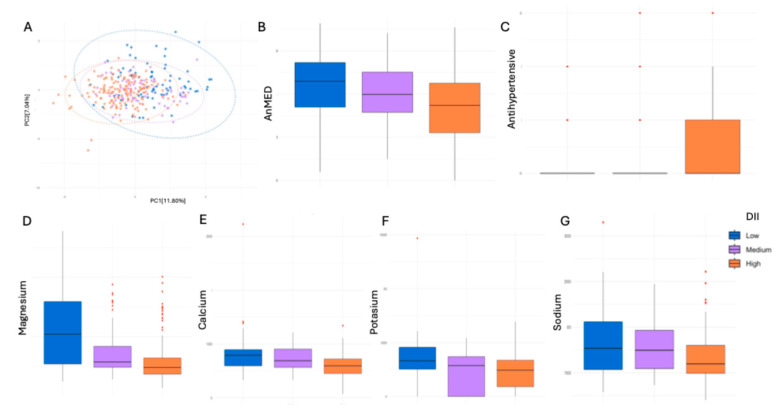
(**A**) PCA representing dietary variability according to DII categorization. (**B**) Adherence to the AnMED diet in relation to DII. (**C**) Association between DII and AHT use. (**D**–**G**) Relationship between AHT use and key micronutrient intake: magnesium (**D**), calcium (**E**), potassium (**F**), and sodium (**G**).

**Table 1 nutrients-17-01570-t001:** MED, DASH, and ANMED diet guidelines.

	MED	DASH	AnMED
White meat	Preferential consumption over red meat. Non-consumption guidelines.	2 or less per day.	3 servings per week.
Red meat	Less than 1 per day.	Not recommended.	Not allowed.
Legumes	3 or more servings per week.	4–5 per week.	3 or more servings per week.
Refined grains	No guidelines.	7–8 servings per day, of which at least half are whole grains.	Not allowed.
Vegetables	2 or more servings per day.	4–5 servings per day.	Distinguishes between green vegetables, non-green vegetables, and other vegetables. A quantity of 100 g of each vegetable group daily.
Fruits	3 or more servings per day.	4–5 servings per day.	Differentiates between enzymatic, antioxidant, and other fruits. One piece of each type per day.
Fish	3 or more times per week.	2 or less per day.	Distinguishes between blue and white fish. Three or more servings of each type per week.
Fats and oils	EVOO primary oil use. Four or more tablespoons per day (60 mL).	2–3 per day.	3 EVOO tablespoons per day (approximately 45 mL).
Nuts	3 or more times per week.	4–5 per week.	1 serving per day.
Dairy products	No guidelines.	2–3 servings per day of low-fat dairy products.	Cow milk not allowed; cured cheese or goat/sheep cheese and natural yogurt or kefir are preferred.
Coffee	No guidelines.	No guidelines.	Maximum of 2 cups per day.
Alcohol	7 glasses of red wine per week.	≤1 drink for women. ≤2 drinks for men.	Not allowed.
Sugar and pastries	Less than 2 servings per week.	5 or less per week.	Not allowed.
Saturated fats products	Less than 7 per week.	Not recommended.	Not allowed.
Sweet or carbonated beverages	Less than 1 per day.	Not recommended.	Not allowed.
Turmeric	No guidelines.	No guidelines.	5 g per day.

**Table 2 nutrients-17-01570-t002:** People with high adherence to different diets studied. Results in percentages.

Treated with	*n*	Mean Age	DII	MED	DASH	AnMED
CV treatments	150	69.37 ± 9.44	−0.29 ± 1.31	70.47	5.37	7.38
AHT	37	72.70 ± 8.57	−0.20 ± 1.04	77.78	0	5.56
ADM	19	63.50 ± 9.97	−0.62 ± 1.08	78.95	5.26	10.53
LLT	34	67.90 ± 9.05	−0.31 ± 1.39	61.76	8.82	5.88
AHT and ADM	2	71.50 ± 10.6	−0.14 ± 1.68	100	0	0
AHT and LLT	34	72.20 ± 8.33	0.08 ± 1.63	61.76	5.88	2.94
LLT and ADM	13	62.20 ± 8.06	−1.09 ± 0.92	76.92	15.39	30.77
AHT, ADM, and LLT	11	72.50 ± 8.59	−0.13 ± 1.24	72.73	0	0
Other drugs	105	65.60 ± 9.32	−0.56 ± 1.18	65.09	2.83	6.60
Non-drugs consumption	49	57.70 ± 3.19	−1.77 ± 0.75	73.47	10.20	2.04

**Table 3 nutrients-17-01570-t003:** Macronutrient intakes.

	CV Treatments	Other Drugs	Non-Drugs	*p*-Value	*p*-Adjust
Total proteins	11.9%	12.2%	13.1%	0.778	1.000
Total carbohydrates	30.6%	32.9%	35.3%	0.518	1.000
Total fats	57.6%	54.9%	51.6%	**0.003**	**0.004 ****
SFA	17.95 ± 6.70 (g)	17.39 ± 6.98 (g)	13.09 ± 3.32 (g)	**<0.001**	**<0.001** *******
PUFA	12.47 ± 4.53 (g)	12.05 ± 4.30 (g)	10.18 ± 2.49 (g)	**0.005**	**0.043 ***
MUFA	72.48 ± 34.98 (g)	66.90 ± 33.57 (g)	52.32 ± 16.25 (g)	**0.001**	**0.011 ***
Lauric acid	0.20 ± 0.17 (g)	0.21 ± 0.19 (g)	0.11 ± 0.13 (g)	**<0.001**	**0.001 ****
Miristic acid	0.81 ± 0.58 (g)	0.87 ± 0.62 (g)	0.54 ± 0.39 (g)	**0.002**	**0.017 ***
Palmitic acid	3.74 ± 1.54 (g)	4.04 ± 1.63 (g)	3.21 ± 1.01 (g)	**0.010**	0.091
Stearic acid	3.66 ± 1.35 (g)	3.58 ± 1.38 (g)	2.71 ± 0.66 (g)	**<0.001**	**<0.001 *****
Oleic acid	65.77 ± 31.59 (g)	60.84 ± 30.22 (g)	47.66 ± 14.78 (g)	**0.001**	**0.009 ****

SFA: Saturated Fat Acid. PUFA: Polyunsaturated Fat Acid. MUFA: Monounsaturated Fat Acid. Significant *p*-values are indicated in bold. *: *p*-value < 0.05; **: *p*-value < 0.01; ***: *p*-value < 0.001.

**Table 4 nutrients-17-01570-t004:** Predictive model for the use of AHT drugs based on DII.

DII	Predicted Probability (Fit)	Lower Bound	Upper Bound
−2	0.168	0.061	0.275
−1	0.311	0.235	0.387
0	0.454	0.373	0.535
1	0.597	0.480	0.713
2	0.739 4	0.575	0.904

**Table 5 nutrients-17-01570-t005:** Dietary patterns and micronutrient intakes.

	MED	DASH	AnMED
Calcium	X	X	V
Magnesium	X	X	V
Potasium	X	V	V
Sodium	X	V	V

## Data Availability

Data described in the manuscript, code book, and analytic code will be made available upon request to corresponding author.

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
