# Peer review of "Relationship Between Dietary Inflammatory Index, Diets, and Cardiovascular Medication"

_nutrients, 2025, doi:10.3390/nu17091570_

Round 1
Reviewer 1 Report
Comments and Suggestions for Authors
Thank you for the opportunity to review this study entitled “Relationship between dietary inflammatory index, diets and cardiovascular medication” (nutrients-3610594).
The paper investigated the association between adherence to healthy dietary patterns and Cardiovascular treatments. The research involved 304 patients.
The manuscript addresses a relevant and timely topic; overall, it represents a valuable contribution to the literature. A major strength of the study lies in its comprehensive approach, combining dietary pattern adherence, dietary inflammatory potential, and pharmacological data to explore cardiovascular risk. However, there are several aspects that would benefit from further clarification or refinement before publication. Below are some suggestions aimed at enhancing the clarity and completeness of the manuscript:
- Abstract: The final sample size is not clearly reported in the abstract. Additionally, it would be helpful to include more descriptive information (e.g., mean age, sex distribution) to provide a clearer overview of the study population.
- Keywords: Please ensure that the keywords are listed in alphabetical order, in accordance with academic conventions.
- Introduction: In the sentence “In Table 1 we can observe the differences between the three dietary models”, consider adopting a more formal, impersonal tone. For example: “Table 1 presents the differences between the three dietary models.”
- Introduction: Expressions such as “it is imperative to investigate” may sound overly deterministic. Consider using more neutral phrasing, such as “it is important to investigate” or “it may be useful to explore”.
- Introduction: The authors could consider explicitly formulating one or more hypotheses based on the literature reviewed.
- Methods and Results: These sections are clearly written and well-structured. No major issues were identified.
- Discussion: The discussion includes interesting associations between higher DII scores and the use of beta-blockers and ARBs. However, further interpretation of these findings would be valuable, particularly in light of the differing mechanisms of action and patient profiles across antihypertensive drug classes. The authors may consider briefly discussing the plausibility and clinical relevance of these associations.
- Discussion: While the authors acknowledge the modest R² value of their predictive model, it would be helpful to further elaborate on the limitations of the cross-sectional design in inferring causality between the dietary inflammatory index (DII) and antihypertensive drug use. This would improve the interpretability of the findings and their relevance for clinical or preventive strategies.
- Conclusions: The conclusions are appropriate and supported by the results. However, it would be advisable to clearly state that the findings are preliminary and not sufficient to inform clinical decisions based solely on DII scores.
Best regards
Author Response
Thank you for the opportunity to review this study entitled “Relationship between dietary inflammatory index, diets and cardiovascular medication” (nutrients-3610594).
Thank you for taking the time to review our manuscript and provide valuable feedback. Your comments have helped us improve the clarity and quality of our work, and we appreciate your thoughtful suggestions.
The paper investigated the association between adherence to healthy dietary patterns and Cardiovascular treatments. The research involved 304 patients.
The manuscript addresses a relevant and timely topic; overall, it represents a valuable contribution to the literature. A major strength of the study lies in its comprehensive approach, combining dietary pattern adherence, dietary inflammatory potential, and pharmacological data to explore cardiovascular risk. However, there are several aspects that would benefit from further clarification or refinement before publication. Below are some suggestions aimed at enhancing the clarity and completeness of the manuscript:
Abstract: The final sample size is not clearly reported in the abstract. Additionally, it would be helpful to include more descriptive information (e.g., mean age, sex distribution) to provide a clearer overview of the study population.
Thank you for your helpful suggestion. In response to your comment, we have updated the Results section of the abstract (lines 20-21) to report the final sample size, sex distribution, and mean age of the participants, as follows: “304 were included in the final analysis (88.48% female, mean age: 66.16 ± 9.59 years).”
Keywords: Please ensure that the keywords are listed in alphabetical order, in accordance with academic conventions.
We appreciate your suggestion and have now listed the keywords in alphabetical order.
Introduction: In the sentence “In Table 1 we can observe the differences between the three dietary models”, consider adopting a more formal, impersonal tone. For example: “Table 1 presents the differences between the three dietary models.”
As suggested, we have revised the sentence for clarity. It now reads: 'Table 1 presents the differences between the three dietary models.’ In line 101
Introduction: Expressions such as “it is imperative to investigate” may sound overly deterministic. Consider using more neutral phrasing, such as “it is important to investigate” or “it may be useful to explore”.
Thank you for your suggestion. We have revised the sentence to adopt a more neutral tone. The phrase has been replaced with: “it is important to investigate.” In line 139
Introduction: The authors could consider explicitly formulating one or more hypotheses based on the literature reviewed.
We appreciate the reviewer’s suggestion. In response, we have revised the final paragraph of the Introduction to explicitly formulate our study hypotheses, in line with the literature previously discussed.
The following sentences have been added in lines 148–151: “Based on the reviewed literature, we hypothesize that individuals with higher DII scores are more likely to require antihypertensive pharmacological treatment. Furthermore, we expect that pro-inflammatory dietary patterns may be indirectly associated with a greater likelihood or intensity of AHT drug use.”
Methods and Results: These sections are clearly written and well-structured. No major issues were identified.
Discussion: The discussion includes interesting associations between higher DII scores and the use of beta-blockers and ARBs. However, further interpretation of these findings would be valuable, particularly in light of the differing mechanisms of action and patient profiles across antihypertensive drug classes. The authors may consider briefly discussing the plausibility and clinical relevance of these associations.
We appreciate the reviewer’s insightful comment. In response, we have expanded the Discussion section to include a more detailed interpretation of the observed associations. Specifically, we now address the clinical plausibility of the relationship between higher DII scores and the increased use of beta-blockers and ARBs. We highlight that these antihypertensive drug classes are among the most frequently prescribed in patients with complex cardiovascular risk profiles, often characterized by chronic low-grade inflammation and poor dietary habits. We also note that while the differing mechanisms of action are acknowledged, the associations observed are more likely to reflect the inflammatory and metabolic context of these patients rather than a direct pharmacological effect. These additions aim to clarify the clinical relevance of our findings and align with the reviewer’s suggestion. The revised text can be found in the Discussion section, lines 397-403; “These associations may reflect the clinical situation in which beta-blockers and ARBs are among the most prescribed antihypertensive drugs, especially in patients with multiple comorbidities, such as obesity, diabetes, or metabolic syndrome, conditions often accompanied by low-grade chronic inflammation and suboptimal dietary patterns. Given their differing mechanisms of action, it is plausible that their association with higher DII scores reflects not a pharmacological effect per se, but rather the inflammatory and dietary profile of the patients who require them.”
Discussion: While the authors acknowledge the modest R² value of their predictive model, it would be helpful to further elaborate on the limitations of the cross-sectional design in inferring causality between the dietary inflammatory index (DII) and antihypertensive drug use. This would improve the interpretability of the findings and their relevance for clinical or preventive strategies.
We thank the reviewer for this suggestion. We have now expanded the Discussion to explicitly address the limitations of using a cross-sectional design in the context of predictive modeling. Specifically, we clarify that while our model suggests a statistical association between higher DII scores and the probability of AHT drug use, the nature of the study design precludes causal inference. We now discuss how the temporal ambiguity inherent in cross-sectional analyses limits the ability to determine whether pro-inflammatory dietary patterns contribute to increased drug use, or vice versa.
The following information was added in lines 381–385 of the revised manuscript: “Moreover, it is important to acknowledge that our study's cross-sectional design limits the ability to draw causal conclusions. The temporal direction of the association between DII and antihypertensive treatment cannot be firmly established, meaning we cannot exclude the possibility that individuals receiving pharmacological treatment may also have modified their diets in response to their diagnosis.”
Conclusions: The conclusions are appropriate and supported by the results. However, it would be advisable to clearly state that the findings are preliminary and not sufficient to inform clinical decisions based solely on DII scores.
We thank the reviewer for this observation. In response, we have clarified the scope of our findings in the Conclusions section. The following information has been added in lines 527–528: “The present findings are preliminary and should not be interpreted as sufficient to guide clinical decisions based solely on DII scores.”This change may reinforces the exploratory nature of our results and avoids overinterpretation.
Reviewer 2 Report
Comments and Suggestions for Authors
A cross-sectional study conducted on a convenience sample from an unspecified population. There are several points that require improvement.
- In the abstract the authors indicate 468 participants but failed to indicate the actual number of people examined.
- The introduction begins with the statement that hypertension is the main cardiovascular risk factor, without providing data to support this statement. Immediately after, the authors recall that cardiovascular risk is associated with several factors and mention inflammation. This beginning is a bit confusing. Since the study focuses on inflammation, it would be better to define inflammation, say what diseases it is associated with, and make it clear that this is important for cardiovascular risk.
- In the second part of the Introduction, the authors indicate some dietary treatments that are thought to be able to reduce inflammation. In citing these models, the authors should indicate the authors of each guideline.
- On line 106, the authors introduce Dietary Inflammatory Index (DII). Again, they should indicate the authors of the metric.
- Line 146. The authors should explain what the DADER methodology is.
- The authors should state how they accessed the health data (electronic medical records and dispensing records of the participants).
- Line 148. Authors should explain what the Food Frequency Questionnaire (FFQ) is and what the PREDIMED group is. They should indicate how the questionnaire is made, provide an example of the questions, indicate the authors of the questionnaire, and when the validation study was done (if any).
- The authors should say whether their survey of dietary habits had any time reference
- At line 208 the authors indicate the sample as "patients"; a few lines below, however, they state that one sixth of the participants had no diagnosis or therapy. Consequently, the sample is not composed of patients, but of two categories. It is necessary to clarify.
- In Methods, the authors state that “In the present study, the group of CV treatment has been considered to be those patients in chronic treatment with AHT, lipid-lowering treatment (LLT) or antidiabetics (ADM)”.
- In Table 2, the cases of CV treatment are 150, with considerable dispersion among the different drug combinations. There is a group of 105 people classified as "Other drugs". Can the authors exclude that all the CV treated patients did not also take other drugs?
- Table 3. SFA, PUFA, MUFA, acronyms not previously explained.
- In the manuscript, the "Limitations" section is completely absent. The authors must reflect on some points: the number of observations is very small, the subdivision by type of diet and type of treatment determines numerous comparisons between very small groups, with the risk of producing falsely significant results. There was no sampling, the convenience sample is unbalanced between elderly people with diseases and healthy young people. The cross-sectional nature of the study prevents giving causal value to the associations found.
- At line 426 the authors noted that "there are some results of our work that do not coincide with those reported in the literature". They should elaborate on this point and clarify that the result may arise from the weakness of the research model conducted.
- In Conclusions, the authors state that “The DII is a robust tool for evaluating the inflammatory potential of diets and has significant implications for understanding and managing diet-related inflammation and chronic diseases”. This statement is not derived from the results of the study. The authors should limit themselves to commenting on what they observed.
- At line 457 the authors conclude that "diets... are effective in lowering blood pressure and improving overall CV health." This conclusion is not admissible, given the nature of the study. At most, the authors could observe that the diet is associated with...
Author Response
A cross-sectional study conducted on a convenience sample from an unspecified population. There are several points that require improvement.
In the abstract the authors indicate 468 participants but failed to indicate the actual number of people examined.
We appreciate your observation. To clarify the discrepancy between the number of participants recruited and those analyzed, we have revised the abstract’s Results section (lines 20–22) to explicitly indicate the final number of participants included in the study: “Of 468 participants initially recruited, 304 were included in the final analysis.” We have also added a brief description of the sex distribution and mean age of the final sample to improve the reader’s understanding of the study population.
The introduction begins with the statement that hypertension is the main cardiovascular risk factor, without providing data to support this statement. Immediately after, the authors recall that cardiovascular risk is associated with several factors and mention inflammation. This beginning is a bit confusing. Since the study focuses on inflammation, it would be better to define inflammation, say what diseases it is associated with, and make it clear that this is important for cardiovascular risk.
Thank you for your comment. Regarding the observation about the initial statement that HT is the main cardiovascular risk factor, we would like to point out that this information is supported by reference [1], an article previously published in this same journal. However, we understand that the connection between HT and other risk factors may not have been sufficiently clear. To address this observation and provide better clarity, we have revised the introduction and made an adjustment in lines 40-44. "Chronic low-grade inflammation plays a key role in the pathogenesis of diseases that exhibit shared CV risk factors, including HT, dyslipidemia, and diabetes mellitus (DM), promoting endothelial dysfunction, oxidative stress, and arterial stiffness, which are key factors in the onset of HT and cardiovascular disease."
We hope this adjustment clarifies the relationship between HT and inflammation and contributes to a better understanding of the context of our research.
In the second part of the Introduction, the authors indicate some dietary treatments that are thought to be able to reduce inflammation. In citing these models, the authors should indicate the authors of each guideline.
We appreciate your insightful feedback regarding the need to attribute the authorship of the dietary models mentioned in the Introduction. To address this point and enhance the clarity of our manuscript, we have modified the text as follows:
Line 71: We have included the attribution, stating: "The Mediterranean diet (MED), described by the PREDIMED group…"
Lines 79-80: For the DASH diet, we added: "The Dietary Approaches to Stop Hypertension (DASH) regimen, developed by the National Institutes of Health,..."
Lines 88-89: Regarding the AnMED diet, we now specify: "The Anti-inflammatory diet (AnMED), as described by Sala-Climent et al., is characterized as the most restrictive one..."
On line 106, the authors introduce Dietary Inflammatory Index (DII). Again, they should indicate the authors of the metric.
As in your previous comment, we have added the authorship of the DII metric, now in line 125: ‘...as described by Shivappa et al.,’
Line 146. The authors should explain what the DADER methodology is.
Thank for your observation. We agree that the DADER methodology, while widely recognized within the pharmaceutical domain, may not be as familiar to readers from other fields. To address this, we have clarified the procedure in the text. Specifically, we added the following explanation in lines 170-175: "Drug information was documented following a systematic approach for pharmacotherapeutic follow-up aimed at optimizing medication use, preventing medication-related problems, and improving health outcomes. This method involves the collection and analysis of pharmacotherapeutic data, identification of potential issues, and, if necessary, proposing interventions to ensure treatment efficacy and safety."
The authors should state how they accessed the health data (electronic medical records and dispensing records of the participants).
Thank you for your valuable observation. We agree that clarification on how health data were accessed is essential. In response to your comment, we have specified in lines 167-168 that: “…Data were obtained through personal interviews and from the electronic medical records and dispensing data of the participants”.
Line 148. Authors should explain what the Food Frequency Questionnaire (FFQ) is and what the PREDIMED group is. They should indicate how the questionnaire is made, provide an example of the questions, indicate the authors of the questionnaire, and when the validation study was done (if any).
Thank you for your valuable feedback. Regarding your suggestion to elaborate on the Food Frequency Questionnaire (FFQ) and the PREDIMED group, we would like to address this comment in the context of the established standards of the field.
The FFQ is a widely recognized tool in nutritional epidemiology, extensively validated and used in numerous studies, including those published in Nutrients. In acknowledgment of your suggestion and to ensure clarity for a broader audience, we have revised the manuscript accordingly. Specifically, we have added that the FFQ used in our study is a validated questionnaire consisting of 161 items, which participants completed by reporting their weekly or monthly intake for each food item. Daily consumption (in grams per day) was calculated by multiplying the reported frequency of consumption by the standard portion sizes provided for each item (lines 176-181). “… the validated Food Frequency Questionnaire (FFQ) utilized by the PREDIMED group was initially employed to collected data regarding food consumption. The questionnaire comprised 161 items, which participants were asked to complete, indicating their weekly or monthly intake of each item. Daily consumption for each item in grams/day was estimated by multiplying the reported consumption frequency by the average daily intake.”
These revisions aim to strike a balance between providing essential details and maintaining the focus of the manuscript, without overextending into well-established methodological descriptions that may be redundant for the primary audience of this journal.
We trust that these adjustments address your concerns while ensuring the manuscript remains aligned with the standards of similar publications in Nutrients. Thank you for your constructive input, which has helped refine the clarity of our work.
The authors should say whether their survey of dietary habits had any time reference
We thank the reviewer for the observation. We would like to clarify that the time frame of the dietary survey is indeed specified in the Methods section. As stated in line 154, the study was conducted over a five-month period, from April to November 2024, during which participants were recruited, and data were collected. This period reflects the reference timeframe for all survey-based variables, including dietary habits.
At line 208 the authors indicate the sample as "patients"; a few lines below, however, they state that one sixth of the participants had no diagnosis or therapy. Consequently, the sample is not composed of patients, but of two categories. It is necessary to clarify.
Thank you for your observation. To improve clarity and accuracy, we have replaced the term “patients” with “participants” now in line 240 throughout the manuscript, as the sample includes both individuals with and without a formal diagnosis or therapy.
In Methods, the authors state that “In the present study, the group of CV treatment has been considered to be those patients in chronic treatment with AHT, lipid-lowering treatment (LLT) or antidiabetics (ADM)”.
In Table 2, the cases of CV treatment are 150, with considerable dispersion among the different drug combinations. There is a group of 105 people classified as "Other drugs". Can the authors exclude that all the CV treated patients did not also take other drugs?
We thank the reviewer for this observation. However, we would like to clarify that the aim of the present study is not to isolate individuals based on exclusive pharmacological profiles, but rather to compare those receiving CV treatment—regardless of potential comorbidities or additional medications—with individuals who may be undergoing treatment for other conditions but not for CV risk, and with a healthy, untreated population. Therefore, while some participants in the CV treatment group may indeed take other non-CV medications, this does not conflict with our study design or objectives, which focus on the impact of CV-related treatment status on inflammatory dietary patterns.
Table 3. SFA, PUFA, MUFA, acronyms not previously explained.
We appreciate your observation. The acronyms SFA (saturated fatty acids), MUFA (monounsaturated fatty acids), and PUFA (polyunsaturated fatty acids) have now been defined in the footnote of Table 3 (line 265), as well as included in the list of abbreviations to enhance clarity for the reader.
In the manuscript, the "Limitations" section is completely absent. The authors must reflect on some points: the number of observations is very small, the subdivision by type of diet and type of treatment determines numerous comparisons between very small groups, with the risk of producing falsely significant results. There was no sampling, the convenience sample is unbalanced between elderly people with diseases and healthy young people. The cross-sectional nature of the study prevents giving causal value to the associations found.
We appreciate the reviewer’s input, which has strengthened the rigor and transparency of our manuscript. In response, we have now included a dedicated “Limitations” section at the end of the Discussion (lines 514–525), where we explicitly address the key concerns raised: “This study has several limitations that must be acknowledged. Firstly, the cross-sectional design prevents the establishment of causal relationships between DII and the use of AHT drugs. Secondly, the sample size (n = 304), although sufficient for detecting statistically significant associations, limits the generalizability of the findings. Thirdly, the use of a convenience sample, with a natural imbalance between older individuals with CV diseases and younger, healthier participants, may have introduced selection bias. However, a linear regression analysis showed no significant relationship between age and DII (p = 0.164), supporting the inclusion of all participant groups. Additionally, the subdivision by dietary adherence and treatment type led to relatively small subgroups, which could increase the risk of type I errors due to multiple comparisons. These limitations highlight the need for future longitudinal studies with larger, randomized populations to validate our findings.”
At line 426 the authors noted that "there are some results of our work that do not coincide with those reported in the literature". They should elaborate on this point and clarify that the result may arise from the weakness of the research model conducted.
We appreciate yours observation regarding this critical point. In response, we have revised the discussion section to include a more detailed explanation of the potential reasons for the observed discrepancies. Lines 487-491: “This discrepancy may arise from the limitations of the research model employed in this study. The cross-sectional nature of the study does not account for variations in sodium consumption over time or the impact of external factors such as food availability, preparation methods, and cultural influences.”
In Conclusions, the authors state that “The DII is a robust tool for evaluating the inflammatory potential of diets and has significant implications for understanding and managing diet-related inflammation and chronic diseases”. This statement is not derived from the results of the study. The authors should limit themselves to commenting on what they observed.
We agree with the reviewer that the statement may have conveyed more certainty than warranted by our data. We have modified the sentence to better reflect the scope of our findings. In line 529, the sentence now reads: “The DII is a widely used tool for evaluating the inflammatory potential of diets and has potential implications for understanding and managing diet-related inflammation and chronic diseases.”
At line 457 the authors conclude that "diets... are effective in lowering blood pressure and improving overall CV health." This conclusion is not admissible, given the nature of the study. At most, the authors could observe that the diet is associated with...
We thank the reviewer for this important observation. We acknowledge the limitation of our cross-sectional design and agree that causal conclusions cannot be drawn. Accordingly, we have rephrased the sentence now in line 535 to reflect the observational nature of the findings.
The revised sentence now reads (lines X–X): “These diets, rich in fruits, vegetables, whole grains, low-fat dairy, and low in sodium, saturated fats, and refined grains, are associated with lower blood pressure levels and improved overall CV health.”
Thank you for taking the time to review our manuscript and provide valuable feedback. Your comments have helped us improve the clarity and quality of our work, and we appreciate your thoughtful suggestions.
Reviewer 3 Report
Comments and Suggestions for Authors
This study evaluated the link between healthy dietary patterns and cardiovascular medication use. It assessed participants' adherence to the MED, DASH, and AnMED diets via food frequency questionnaires, calculated the DII, and examined its relationship with the use of antihypertensive, lipid-regulating, and antidiabetic medications. The findings indicate that DII effectively measures a diet's inflammatory potential. Low DII diets like AnMED may enhance cardiovascular health, reduce reliance on antihypertensive drugs, and provide a scientific basis for dietary interventions in preventing and managing cardiovascular diseases. The article is well-structured with innovative conclusions, but could be improved in the following aspects:
Introduction
1. The introduction devoted excessive space to antihypertensive drug side effects, a concise summary would suffice.
2. To aid reader comprehension of the core metric, the definition, calculation method of DII, and its current applications in cardiovascular research should be included.
3. The connection between paragraphs in the introduction should be strengthened, such as the correlation between MED, DASH, AnMED diets and the DII index.
Methods
1. In "2.1 Study Design", the specific sample size, gender ratio, and criteria for "high adherence" should be clarified, along with a more detailed explanation of exclusion criteria.
2. The specific formula or reference for DII calculation should be provided to enhance reproducibility.
Results
1. Tables should follow the three-line format, and the number of decimal places in data within tables should be standardized.
2. The legends for Figures 2 and 3 need to be clearer.
3. Standard deviations or confidence intervals should be added to the percentage data in Table 2 to prevent potential misunderstandings.
4. In "3.3 Predictive Model", possible reasons for the low explanatory power of DII (R² = 7.6%) should be explored.
Discussion
1. A mechanistic analysis of why AnMED diet performed better in micronutrient intake should be included.
2. The limitations of the study should be clearly stated.
3. A comparison of the mechanistic differences between AnMED and other diets is necessary.
Conclusion
1. The synergistic relationship between dietary interventions and drug therapy should be emphasized, and absolute statements suggesting that one can replace the other should be avoided.
2. Directions for future research should be proposed.
Others
1. The list of abbreviations should be sorted alphabetically.
2. The method of data acquisition should be specified to align with open science practices.
Author Response
Thank you for taking the time to review our manuscript and provide valuable feedback. Your comments have helped us improve the clarity and quality of our work, and we appreciate your thoughtful suggestions.
- The introduction devoted excessive space to antihypertensive drug side effects, a concise summary would suffice.
We appreciate the reviewer’s observation and agree that a more concise description improves the clarity and focus of the introduction. Accordingly, we have reduced the level of detail regarding antihypertensive side effects, while preserving the essential message about their impact on adherence and quality of life. The revised version appears in the Introduction section, lines 51-52, and now reads as follows: “...such as fatigue, dizziness, electrolyte imbalances, or persistent cough, which can contribute adherence and quality of life [5–7].”
- To aid reader comprehension of the core metric, the definition, calculation method of DII, and its current applications in cardiovascular research should be included.
Thank you for your comment. The calculation method of the DII is explained in detail in Section 2.3, “Statistical Inference.” However, we acknowledge that it was not explicitly stated at the beginning of the paragraph that the procedure described refers to the DII. To improve clarity, we have added a sentence (The following procedure describes the calculation of the DII. Line 202) at the beginning of the paragraph to indicate that the following description corresponds to the calculation of the DII.
Regarding the current applications of the DII in cardiovascular research, this information is already addressed in the Introduction section, where we discuss its association with cardiometabolic risk factors, blood pressure, and arterial stiffness (lines 125-137).
- The connection between paragraphs in the introduction should be strengthened, such as the correlation between MED, DASH, AnMED diets and the DII index.
Thank you for your valuable feedback. In response to your comment, we have revised the introduction to better articulate the connection between the dietary patterns (MED, DASH, AnMED) and their inflammatory potential as captured by the Dietary Inflammatory Index (DII). Specifically, we included transitional phrases and added clarifying sentences (see lines 95-96 “Importantly, these three dietary models differ in their inflammatory potential, which can be evaluated using the Dietary Inflammatory Index (DII).” and 122-124 “As such, their varying levels of adherence may result in different DII scores, with AnMED generally associated with lower (more favorable) DII values.”) to explicitly highlight how the inflammatory profiles of these diets differ and how this is reflected in their corresponding DII scores. We believe this enhancement improves the flow and clarity of the rationale for our investigation.
Methods
- In "2.1 Study Design", the specific sample size, gender ratio, and criteria for "high adherence" should be clarified, along with a more detailed explanation of exclusion criteria.
Thank you for your valuable comment. The criteria for "high adherence" to the dietary models are detailed in Section 2.2 “Participants and Data Collection,” lines 187-191, where we describe the scoring thresholds for the AnMED, MED, and DASH diets. However, to improve clarity, we have added further detail regarding the exclusion criteria to Section 2.1. Specifically, we have now indicated that participants without medication or dietary records were excluded from the analysis with the next sentence (lines 161-162): “Additionally, participants without a record of prescribed medications and/or dietary intake data were excluded from the study.”
In response to your comment regarding the need to clarify the gender ratio of the study population, we have included this information in the Results section (line 242) as follows: “A total of 304 patients (88.48% female) were finally included...”
- The specific formula or reference for DII calculation should be provided to enhance reproducibility.
Thank you for your comment. The calculation method of the DII is explained in detail in Section 2.3, “Statistical Inference” (lines 202-211), and the references who explain DII calculation are 25, 27, 32, 33, 35 and 36. All of them were derived from the same researchers group who design this evaluation.
Results
- Tables should follow the three-line format, and the number of decimal places in data within tables should be standardized.
Thank you for your observation. In response, we have modified Table 2 (line 251) to adhere to the three-line format, eliminating horizontal lines within the same group to ensure a cleaner presentation. Additionally, we have standardized the number of decimal places in Tables 2, 3, and 4, (lines 251, 264 and 312 respectively) rounding the data as appropriate to maintain consistency throughout the manuscript. We hope these changes address your concerns and improve the clarity and presentation of the tables.
- The legends for Figures 2 and 3 need to be clearer.
In response to your suggestion, we have clarified the legends for Figures 2 and 3 to ensure they are more descriptive and easier to interpret. Below are the revised figure legends:
Lines 285-289: “Figure 2. (A) Relationship between DII and adherence to MED diet. (B) DII and adherence to DASH diet. (C) DII and adherence to AnMED diet. The broken line rep-resents the cut-off point between high and low adherence to dietary patterns. (D) DII distribution according to the patient's medication profile. (E) Berry consumption in relation to the patient's medication profile. (F) Magnesium intake based on the patient's medication profile”
Lines 297-299: “Figure 3. (A) PCA of dietary variability based on AHT treatments. (B) DII in relation to AHT drug use. (C) Magnesium intake in relation to AHT drug use. (D) Berries intake in relation to AHT drug use.”
- Standard deviations or confidence intervals should be added to the percentage data in Table 2 to prevent potential misunderstandings.
Thank you for your comment. We understand your concern regarding the interpretation of the percentage data presented in Table 2. However, these percentages represent the proportion of individuals within each treatment group who achieved high adherence to the respective dietary patterns, based on the criteria established in the methodology (≥ 9/14 points for MED and AnMED, and ≥ 7.5/10 for DASH).
As these values are simply percentages calculated based on the number of participants meeting the adherence criteria, we believe that adding standard deviations or confidence intervals is not necessary in this case, as they do not represent a distribution of data but rather a descriptive summary of the adherence status within each treatment group.
We hope this clarifies the situation, and we are pleased to provide any further information if needed.
- In "3.3 Predictive Model", possible reasons for the low explanatory power of DII (R² = 7.6%) should be explored.
We appreciate the reviewer’s comment and have now added a brief discussion of potential factors that may account for the modest R² value of our predictive model. In particular, we note that AHT treatment is a multifactorial outcome, influenced by a variety of clinical, behavioral, and sociodemographic variables beyond dietary inflammation. Additionally, we recognize that the DII alone may not fully capture all dietary or lifestyle components relevant to hypertension risk.
The following content was added in lines 376–380 of the revised manuscript: “This modest explanatory power likely reflects the multifactorial nature of antihypertensive drug use, which is influenced by a wide range of variables such as comorbidities, genetic predisposition, lifestyle habits, and access to healthcare. Additionally, the DII, while informative, may not fully account for the complexity of dietary behaviors and their physiological effects.”
Discussion
- A mechanistic analysis of why AnMED diet performed better in micronutrient intake should be included.
Thank you for your constructive feedback. The mechanistic analysis of the AnMED diet's has been addressed in detail in the introduction, specifically in lines 88-124.
To further clarify this important point, we have revised the discussion section by incorporating a succinct mechanistic analysis. The addition has been made after the paragraph discussing the results of micronutrient intake (lines 461–470), with the next information: “The dietary pattern of the AnMED diet, which restricts pro-inflammatory foods such as red meats, processed products, and added sugars, alongside its emphasis on dietary diversity and the inclusion of nutrient-dense groups such as blue and white fish, daily nuts, and categorized fruits and vegetables, explains its benefits in essential micronutrient intake. These restrictions reduce the consumption of calorie dense but nutritionally poor foods, while the inclusion of anti-inflammatory and nutrient-rich components enhances the intake and bioavailability of key nutrients such as magnesium, potassium, calcium, and antioxidants compounds. This synergy highlights the potential of the AnMED diet to improve not only nutrient adequacy but also overall metabolic and inflammatory profiles.” We believe that these adjustments directly address yours concerns and enhance the clarity and complete the discussion. Please let us know if further elaboration is required.
- The limitations of the study should be clearly stated.
Thank you for this important observation. As recommended, we have now added a “Limitations” paragraph to the Discussion section (lines 514–525), where we clearly acknowledge the cross-sectional design of the study, the modest sample size, the unbalanced convenience sampling, and the potential inflation of significance due to subgroup comparisons. We believe this addition enhances the integrity of our study and provides the reader with a more nuanced understanding of our results.
The next paragraph was added: “This study has several limitations that must be acknowledged. Firstly, the cross-sectional design prevents the establishment of causal relationships between DII and the use of AHT drugs. Secondly, the sample size (n = 304), although sufficient for detecting statistically significant associations, limits the generalizability of the findings. Thirdly, the use of a convenience sample, with a natural imbalance between older individuals with CV diseases and younger, healthier participants, may have introduced selection bias. However, a linear regression analysis showed no significant relationship between age and DII (p = 0.164), supporting the inclusion of all participant groups. Additionally, the subdivision by dietary adherence and treatment type led to relatively small subgroups, which could increase the risk of type I errors due to multiple comparisons. These limitations highlight the need for future longitudinal studies with larger, randomized populations to validate our findings.”
- A comparison of the mechanistic differences between AnMED and other diets is necessary.
Thank you for this insightful comment. The mechanistic differences between AnMED and the other dietary patterns are described in the Introduction section, including their nutritional components and physiological pathways influencing CV health (lines 88-112). However, in response to the suggestion, we have reinforced this comparison by adding a summarizing paragraph that explicitly highlights the key mechanistic distinctions between AnMED, MED, and DASH diets to aid reader comprehension in lines 113-124.
“In summary, although all three dietary models contribute to CV health, their mechanisms of action differ. The DASH diet is primarily focused on lowering blood pressure through the reduction of sodium and saturated fats, and the increase of potassium, magnesium, and calcium intake. The traditional MED diet exerts its effects through the high intake of EVOO, polyphenols, and omega-3-rich fish, enhancing lipid profiles, antioxidant status, and gut microbiota. The AnMED, is more restrictive and emphasizes an anti-inflammatory profile by excluding red meats, saturated fats, added sugars, and alcohol, while promoting plant-based diversity and specific food groups with antioxidant and immunomodulatory properties, such as turmeric. These mechanistic distinctions may influence their respective inflammatory potential and impact on cardiometabolic outcomes. As such, their varying levels of adherence may result in different DII scores, with AnMED generally associated with lower (more favorable) DII values.”
Conclusion
- The synergistic relationship between dietary interventions and drug therapy should be emphasized, and absolute statements suggesting that one can replace the other should be avoided.
Thank you for this important suggestion. We have reworded the relevant sentence in the Conclusions section to emphasize that dietary interventions are complementary rather than a substitute for pharmacological treatment. The revised sentence (lines 544–546) now reads: “Rather than replacing pharmacological treatment, anti-inflammatory diets may act as a valuable adjunct in cardiovascular disease management by helping to reduce systemic inflammation.”
This change avoids any implication of exclusivity or substitution and instead highlights the potential synergy between lifestyle and pharmacological strategies.
- Directions for future research should be proposed.
Thank you for your suggestion regarding the importance of outlining directions for future research. We fully agree that this is necessary, particularly considering the cross-sectional nature of our study and its inherent limitations. In response to your comment, we have included this point at the end of the newly added "limitations" paragraph in Discussion section (lines 521–525): “…Additionally, the subdivision by dietary adherence and treatment type led to relatively small subgroups, which could increase the risk of type I errors due to multiple comparisons. These limitations highlight the need for future longitudinal studies with larger, randomized populations to validate our findings.”
Others
- The list of abbreviations should be sorted alphabetically.
We appreciate the reviewer’s observation. We agree that listing abbreviations in alphabetical order improves the clarity and readability of the manuscript. Accordingly, we have reorganized the list of abbreviations in alphabetical order.
Additionally, in response to suggestions from another reviewer, we have included three new abbreviations that are now referenced in the main text. These changes contribute to the overall coherence and completeness of the manuscript.
Now, abbreviations list is the following:
|
ADM |
Antidiabetic medication |
|
AHT |
Antihypertensive treatment |
|
AnMED |
Anti-inflammatory diet |
|
ARBs |
Angiotensin Receptor Blockers |
|
CRP |
C-reactive protein |
|
CV |
Cardiovascular |
|
DASH |
Dietary Approaches to Stop Hypertension |
|
DII |
Dietary Inflammatory Index |
|
DM |
Diabetes Mellitus |
|
DM |
Diabetes Mellitus |
|
EVOO |
Extra virgin olive oil |
|
FFQ |
Food Frequency Questionnaire |
|
HT |
Hypertension |
|
IL-6 |
Interleukin-6 |
|
LLT |
Lipid-lowering therapy |
|
MED |
Mediterranean diet |
|
MUFA |
Monounsaturated Fatty Acids |
|
PCA |
Principal Component Analysis |
|
PUFA |
Polyunsaturated Fatty Acids |
|
SFA |
Saturated Fatty Acids |
|
TNF-α |
Tumor necrosis factor-alpha |
- The method of data acquisition should be specified to align with open science practices.
Thank you for your valuable observation. We agree that clarification on how health data were accessed is essential. In response to your comment, we have specified in lines 167-168 that: “…Data were obtained through personal interviews and from the electronic medical records and dispensing data of the participants”.
Round 2
Reviewer 2 Report
Comments and Suggestions for Authors
The authors have revised the manuscript